# In Vitro One-Pot 3-Hydroxypropanal Production from Cheap C1 and C2 Compounds

**DOI:** 10.3390/ijms23073990

**Published:** 2022-04-03

**Authors:** Su-Bin Ju, Min-Ju Seo, Soo-Jin Yeom

**Affiliations:** 1School of Biological Sciences and Biotechnology, Graduate School, Chonnam National University, Yong-bong-ro 77, Gwangju 61186, Korea; jsb9706@naver.com; 2School of Biological Sciences and Technology, Chonnam National University, Gwangju 61186, Korea; mjseo0207@gmail.com

**Keywords:** deoxyribose-5-phosphate aldolase, methanol dehydrogenase, 3-hydroxypropanal, *Thermotoga maritima*, *Lysinibacillus xylanilyticus*, cascade enzymatic bioconversion

## Abstract

One- or two-carbon (C1 or C2) compounds have been considered attractive substrates because they are inexpensive and abundant. Methanol and ethanol are representative C1 and C2 compounds, which can be used as bio-renewable platform feedstocks for the biotechnological production of value-added natural chemicals. Methanol-derived formaldehyde and ethanol-derived acetaldehyde can be converted to 3-hydroxypropanal (3-HPA) via aldol condensation. 3-HPA is used in food preservation and as a precursor for 3-hydroxypropionic acid and 1,3-propanediol that are starting materials for manufacturing biocompatible plastic and polytrimethylene terephthalate. In this study, 3-HPA was biosynthesized from formaldehyde and acetaldehyde using deoxyribose-5-phosphate aldolase from *Thermotoga maritima* (DERA*_Tma_*) and cloned and expressed in *Escherichia coli* for 3-HPA production. Under optimum conditions, DERA*_Tma_* produced 7 mM 3-HPA from 25 mM substrate (formaldehyde and acetaldehyde) for 60 min with 520 mg/L/h productivity. To demonstrate the one-pot 3-HPA production from methanol and ethanol, we used methanol dehydrogenase from *Lysinibacillus xylanilyticus* (MDH*_Lx_*) and DERA*_Tma_*. One-pot 3-HPA production via aldol condensation of formaldehyde and acetaldehyde from methanol and ethanol, respectively, was investigated under optimized reaction conditions. This is the first report on 3-HPA production from inexpensive alcohol substrates (methanol and ethanol) by cascade reaction using DERA*_Tma_* and MDH*_Lx_*.

## 1. Introduction

One- or two-carbon (C1 or C2) compounds have been utilized as attractive substrates for biotechnological production of valuable compounds because they are inexpensive and can be used as bio-renewable platform feedstock [1,2]. The production of value-added products from cheap C1 or C2 compounds via chemical processes has been extensively studied [3,4]. However, these chemical processes occur under high temperatures and pressures and release toxic by-products [5]. Recently, enzymatic processes using biocatalysts have been developed to substitute chemical synthesis [6,7,8]. Methanol (C1) and ethanol (C2) are major compounds considered next generation substrates that can be converted to natural metabolite products, such as acetaldehyde, formaldehyde, and acetyl-CoA-derived products, in microorganisms via short biosynthetic pathways. Various biocatalysts, such as alcohol oxidase, oxidoreductase, and methanol dehydrogenase (MDH), have been considered to convert methanol and ethanol [9,10]. Among these enzymes, MDH (EC 1.1.1.244) is a key enzyme for alcohol oxidation, which catalyzes alcohol conversion to aldehyde [11]. Previous studies have reported that MDH from *Lysinibacillus xylanilyticus* (MDH*_Lx_*) efficiently converts methanol and ethanol to formaldehyde and acetaldehyde, respectively [12]. Alcohol-derived formaldehyde and acetaldehyde are also renewable feedstock that can be converted to chemical derivatives such as 3-hydroxypropionic acid (3-HP), 3-hydroxypropanal (3-HPA), and 1,3-propanediol (PDO), which are important precursors in the industry [13]. To date, they have been produced from glucose or glycerol using metabolically engineered cells, including those producing glycerol dehydrogenase and aldehyde dehydrogenase enzymes [14,15,16,17,18,19,20,21,22,23,24]. Recently, deoxyribose-5-phosphate aldolase (DERA) from *Thermotoga maritima* (DERA*_Tma_*, E.C 4.1.2.4) has been introduced as an intermediate enzyme to condense formaldehyde and acetaldehyde to 3-HPA for PDO biosynthesis in engineered *Escherichia coli* [25]. In this metabolic pathway, aldol condensation between aldehydes is the key reaction point; however, the typical limitations of metabolic engineering include a lack of enzyme engineering or pathway design optimization. In vitro enzymatic reactions have advantages such as enzyme controllability, pathway design, and increasing product titer and yield; however, metabolic engineering in live cells cannot be controlled [26]. Moreover, optimum reaction conditions for 3-HPA production or pathway optimization by applying other enzymes have not been considered yet.

In this study, we optimized reaction conditions such as pH, temperature, DERA*_Tma_* concentration, and formaldehyde and acetaldehyde concentrations as substrates for 3-HPA production. Moreover, we designed and performed a one-pot cascade reaction for 3-HPA biosynthesis from methanol and ethanol using two enzymes, MDH*_Lx_* and DERA*_Tma_* (Figure 1). To the best of our knowledge, this is the first report on 3-HPA production from methanol and ethanol by enzyme cascade reaction and will be a helpful study for metabolic pathway optimization by applying optimal conditions to target enzymes. Moreover, this study will contribute to the industrial application of DERA-based biocatalysts for the production of high-value compounds from cheap C1 and C2 compounds using green and eco-friendly processes.

## 2. Results and Discussion

### 2.1. Characterization of DERA_Tma_

To obtain an enzyme with high aldol condensation activity, the gene encoding DERA*_Tma_* was cloned and expressed in *E*. *coli*. To confirm the molecular weight of DERA*_Tma_*, the recombinant enzyme was purified from the crude cell extract as a soluble protein using a nickel–nitrilotriacetic acid (Ni^2+^–NTA) column. The purified DERA*_Tma_* enzyme (approximately 29.4 kDa), which converts formaldehyde and acetaldehyde to 3-HPA (Figure 2a), showed a single band in SDS-PAGE (Figure 2b). We used this enzyme to produce 3-HPA from formaldehyde and acetaldehyde. The products were successfully analyzed using high performance liquid chromatography (HPLC) with 3-HPA standards (Appendix A). To optimize the reaction conditions for 3-HPA production, we investigated the effects of pH and temperature on DERA*_Tma_* activity toward formaldehyde and acetaldehyde. The DERA*_Tma_* activity was the highest at pH 7.0 and 40 °C (Figure 2c,d). Interestingly, DERA*_Tma_* exhibited >80% optimum activity at pH 6.0–8.0; however, it decreased under alkaline conditions.

The increasing order of pH values for the optimal activity toward deoxyribose phosphate (DRP) of reported recombinant DERAs from *Thermococcus kodakaraensis* [27]; *Hyperthermus butylicus* [28]; *Pyrobaculum aerophilum* [29], *Yersinia* sp. [30], *Paenibacillus* sp. [31], and *Haloarcula japonica* [32]; *Thermotoga maritima* [29]; *Aciduliprofundum boonei* [33], and *Rhodococcus erythropolis* [34]; *E*. *coli* [29], *Haemophilus influenzae* [35], and *Pseudomonas syringae* [36], were 4.0, 5.5, 6.0, 6.5, 7.0, and 7.5, respectively. The increasing order of temperature values for the optimal activity toward DRP of reported recombinant DERAs from *P*. *syringae* [36]; *H*. *influenzae* [35]; *Yersinia* sp. [30] and *Paenibacillus* sp. [31]; *H*. *japonica* [32]; *R*. *erythropolis* [34]; *H*. *butylicus* [28] and *A*. *boonei* [33]; *T*. *kodakaraensis* [27], were 25, 40, 50, 60, 65, 80, and 90 °C, respectively.

Substrate specificity and kinetic parameters of DERA*_Tma_* toward formaldehyde and acetaldehyde as the substrates were measured (Appendix A). The *k*_cat_ and *K*_m_ of DERA*_Tma_* for formaldehyde and acetaldehyde were 0.49 min^−1^ and 2.41 mM, and 0.51 min^−1^ and 6.95 mM, respectively. The catalytic efficiency (*k*_cat_/*K*_m_, min^−1^ mM) of DERA*_Tma_* for formaldehyde was 2.8-fold higher than that for acetaldehyde, suggesting that DERA*_Tma_* prefers formaldehyde (C1) than acetaldehyde (C2). Zeng et al. investigated the *K*_m_ of DERA*_Tma_* toward formaldehyde and acetaldehyde, which were 2.54 and <1 mM (at pH 7.0 and 30 °C), respectively [25]. However, the *K*_m_ for formaldehyde was similar to that reported previously, but the *K*_m_ value for acetaldehyde was 7-fold higher than that reported previously (at pH 7.0 and 40 °C). Moreover, the reported *K*_m_ values for acetaldehyde in other aldol reactions with sugar substrates for DERA isolated from *Rattus norvegicus* [37], *Lactiplantibacillus plantarum* [38], *E*. *coli* [39], and *Salmonella enterica* [40] were 0.267, 1.1, 1.7, and 3.5 mM, respectively, which were 26.1-, 6.3-, 4.1-, and 1.99-fold lower than that for DERA*_Tma_*, respectively.

### 2.2. Production of 3-HPA from Formaldehyde and Acetaldehyde by DERA_Tma_

To confirm the optimal reaction conditions, we determined the optimal DERA*_Tma_* concentration. The optimal enzyme concentration for 3-HPA production was investigated using 0.28, 0.55, 0.83, 1.11, 1.39, 1.66, 1.94, 2.22, and 2.50 U/mL enzyme toward 10 mM formaldehyde with 20 mM acetaldehyde. The 3-HPA production increased when using <1.66 U/mL enzyme; however, product concentration reached a plateau when using >1.66 U/mL enzyme (Figure 3a). Therefore, we concluded that 1.66 U/mL is the optimal enzyme concentration for 3-HPA production and used it for further studies. To optimize the substrate concentrations, we further investigated 3-HPA production using 0–30 mM formaldehyde or acetaldehyde. The 3-HPA production increased as formaldehyde concentration increased (Figure 3b); however, 3-HPA amount reached a plateau above 15 mM formaldehyde (Figure 3c). Moreover, the optimal acetaldehyde concentration for 3-HPA production was 10 mM. Based on the substrate specificity results, it seems that DERA*_Tma_* prefers formaldehyde as the substrate over acetaldehyde. Thus, we chose 15 mM formaldehyde and 10 mM acetaldehyde for 3-HPA production using DERA*_Tma_*, which yielded 7 mM 3-HPA from 25 mM substrates in 60 min with 520 mg/L/h productivity (Figure 4).

### 2.3. Optimization of the Cascade Reaction with DERA_Tma_ and MDH_Lx_

To investigate 3-HPA production from methanol and ethanol using a cascade reaction (Figure 5a), we optimized the reaction conditions for DERA*_Tma_* and selected MDH*_Lx_* A164F as the best candidate for high concentration methanol oxidation (Appendix A). Previous studies have shown that MDH*_Lx_* showed the highest activity at pH 9.0 and 55 °C in the presence of 5 mM of Mg^2+^ and 3 mM of β-nicotinamide adenine dinucleotide (NAD^+^) [12]. Interestingly, combined DERA*_Tma_* and MDH*_Lx_* A164F showed a maximal activity at pH 7.4 and 45 °C (Figure 5b,c), which are slightly higher than those for the maximal activity of DERA*_Tma_* enzyme alone. To improve the performance of the enzyme cascade reaction, we identified the optimal DERA*_Tma_* and MDH*_Lx_* A164F concentrations for 3-HPA biosynthesis from 1 M methanol and ethanol, which were 1.66 and 0.026 U/mL, respectively (Appendix A). Thus, we used 1.39 U/mL of DERA*_Tma_* and 0.026 U/mL of MDH*_Lx_* A164F for one-pot 3-HPA production from methanol and ethanol.

To obtain the optimal concentrations of methanol and ethanol as substrates, we investigated the enzyme cascade reactions for 3-HPA production using 0.05–1 M methanol in the presence of 1 M ethanol and 0.16–1 M ethanol in the presence of 1 M methanol (Figure 6). The concentration of 3-HPA increased as the methanol concentration increased; however, it reached a plateau when methanol concentration was above 0.8 M. Interestingly, 3-HPA concentration decreased by increasing the ethanol concentration above 0.1 M ethanol. Previous studies have shown that MDH*_Lx_* has higher catalytic efficiency toward ethanol than toward methanol, which can cause interference between the two substrates at the enzyme active site [12], indicating that both enzyme and substrate concentration should be optimized. Finally, we selected 0.8 M methanol and 0.1 M ethanol as suitable substrate concentrations to improve the performance of 3-HPA production.

### 2.4. One-Pot 3-HPA Biosynthesis from Methanol and Ethanol

The optimal reaction conditions for the one-pot 3-HPA biosynthesis using DERA*_Tma_* and MDH*_Lx_* A164F were pH 7.4, 45 °C, 1.66 U/mL of DERA*_Tma_*, 0.026 U/mL of MDH*_Lx_* A164F, 0.8 M of methanol, and 0.15 M of ethanol with 5 mM of Mg^2+^ and 3 mM of NAD^+^. Under these optimal conditions, time-course reactions for 3-HPA biosynthesis were conducted (Figure 7).

Two biocatalysts produced 16.3 mg/L (0.22 mM) of 3-HPA from methanol and ethanol for 8 h with 2.04 mg/L/h productivity. Although the productivity was quite low, producing valuable products from methanol and ethanol as cheap materials without carbon loss and by-product formation is important for value-added material biosynthesis via eco-friendly processes. In addition, MDH and DERA with high activity should be developed using rational approach-based protein engineering or directed evolution in the future. Till date, 3-HPA has been produced as an intermediate by glucose or glycerol fermentation using wild-type strain cells and metabolically engineered cells such as *E*. *coli* [15,21,22,24,25,41], *Lactobacillus* sp. [42,43,44,45,46,47,48], and *Klebsiella pneumoniae* [49,50,51,52,53]. 1,3-butanediol, 3,5-dihydroxypentanal, and 3,5-dihydroxyhexanal [25]; acetate and lactate [15]; 2,3-butanediol [53] were major by-products, respectively, during 3-HP or PDO production via 3-HPA as an intermediate. Moreover, 3-HPA is a toxic intermediate product in bacteria and one of the limitations to producing 3-HPA at a high concentration for further use in the biosynthetic pathway [14].

Thus, 3-HPA has not yet been reported as the main product. However, 3-HPA can be designed as the main product, using enzymatic conversion. Furthermore, methanol and ethanol are considered some of the most useful substrates for obtaining complex chemical compounds. Therefore, these C1 or C2 chemicals have been used to produce valuable products such as polyhydroxyalkanoates, amino acids, organic acids, and other chemical compounds using diverse biocatalysts [1,2]. Thus, processes for the utilization of C1 or C2 compounds as substrates for value-added product manufacturing should be developed in further studies.

## 3. Materials and Methods

### 3.1. Chemicals and Materials

Ethanol, methanol, formaldehyde, acetaldehyde, and NAD^+^ were obtained from Sigma-Aldrich (St. Louis, MO, USA). PrimeSTAR^®^ Max DNA polymerase for cloning the DERA gene was purchased from Takara (Tokyo, Japan). T4 DNA ligase, T4 kinase, and *Dpn*I were obtained from NEB (New England Biolabs; Ipswich, MA, USA). PCR product purification and gel extraction kits were purchased from Promega (Madison, WI, USA). QIA prep Spin mini-prep kit for plasmid preparation was purchased from Qiagen (Hilden, Germany). Ni^2+^-NTA column for protein purification was purchased from GE Healthcare (New York, NY, USA). Oligonucleotides and plasmid sequencing services were provided by Macrogen facility (Seoul, Republic of Korea). All materials for DNA techniques and SDS–PAGE were purchased from Bio-Rad (Hercules, CA, USA). All DNA and protein techniques were carried out following the standard molecular biological protocols. 3-HPA was provided by Korea Research Institute of Bioscience (KRIBB) and used for standard curve using HPLC [54].

### 3.2. Gene Cloning

*E. coli* C2566 (New England Biolabs) and pET-28a(+) carrying kanamycin resistance (Novagen; Madison, WI, USA) were used as host cells and expression vector, respectively. The DERA gene (NCBI accession No. AE000512.1) from *Thermotoga maritima*, comprising 747 nucleotides (248 amino acids), was synthesized, and codon optimization was performed to enhance expression yield in *E*. *coli* by Cosmogenetech facility (Seoul, Republic of Korea). The synthesized gene was cloned into the expression vector pET-28a(+) with His_6_-tag at the N-terminal domain. The DERA gene was amplified from the synthesized gene fragment using PCR and then cleaned up using a PCR purification kit (Promega). The primer sequences used for gene cloning are summarized in Appendix A. The amplified DNA fragments and linearized vector were ligated using Gibson Assembly Master Mix (New England Biolabs). The ligation mixture was transformed into *E*. *coli* C2566 and then plated on Luria Bertani (LB) agar supplemented with 30 μg/mL kanamycin to select the kanamycin-resistant colony. The selected colony was isolated using the plasmid purification kit (Qiagen) and confirmed by DNA sequencing at the Macrogen facility (Seoul, Korea).

### 3.3. Site-Directed Mutagenesis of MDH_Lx_

To improve 3-HPA production, the previously reported MDH*_Lx_* was applied to a one-pot reaction with DERA*_Tma_* [11]. To construct double-variants (K46E-A164F and K318N-A164F) using site-directed mutagenesis, the plasmid template DNA containing the MDH*_Lx_* variants was added to the PCR mixture containing 10 pmol of each primer (Appendix A) with PrimeSTAR^®^ Max [11]. The amplified products were treated with *Dpn*I for 2 h at 37 °C to remove the methylated parental template and hybrid DNA. The treated mixture was transformed into *E*. *coli* C2566. Each colony on the plate was collected and sequenced.

### 3.4. Culture Conditions

Each plasmid, such as the DERA*_Tma_* and MDH*_Lx_* variants, was transformed into *E*. *coli* C2566 using electroporation (Bio-Rad, Hercules, CA, USA). These transformants (10–100 μL) were plated on LB agar with 30 μg/mL kanamycin, incubated at 37 °C overnight (14–16 h), and then inoculated in 3 mL LB broth. Recombinant *E*. *coli* cells harboring DERA*_Tma_* or MDH*_Lx_* were cultivated in a 2-L flask containing 250 mL LB broth supplemented with 30 μg/mL kanamycin at 37 °C and 200 rpm. To induce DERA*_Tma_* expression, isopropyl-β-d-thiogalactopyranoside (IPTG; final concentration: 0.1 mM) was added to the LB broth when the optical density of recombinant cells at 600 nm reached 0.6–0.8 and continuously cultivated at 30 °C with shaking at 200 rpm for overnight. In case of MDH*_Lx_* and its variants, the temperature was reduced to 20 °C after IPTG induction and they were continuously cultivated overnight to induce enzyme expression [11].

### 3.5. DERA_Tma_ and MDH_Lx_ Purification

Recombinant *E*. *coli* cells were harvested by centrifugation at 8660× *g* for 20 min at 4 °C and then stored at −80 °C. Then, they were resuspended in lysis buffer (20 mM Tris–HCl, 0.5 M NaCl, pH 7.0) and disrupted on ice by ultrasonication for 10 min. The disrupted cells were centrifuged at 31,660× *g* for 30 min at 4 °C to remove unbroken cells and cell debris. The supernatant (crude extract) was loaded to a Ni^2+^–NTA column, and the bound protein was eluted with the same buffer containing 250 mM imidazole at 4 °C. The purified enzyme was desalted with a 10 kDa Amicon ultra centrifugal filtration tube (Sigma-Aldrich), dialyzed with 20 mM Kpi buffer (pH 7.0), and confirmed using SDS–PAGE. The protein concentration was measured using the Bradford method.

### 3.6. Enzyme Assay and Kinetic Parameters

One unit (U) enzyme activity was defined as the amount of enzyme required to produce 1 μmol 3-HPA per minute at 40 °C and pH 7.0. The enzyme specific activity was defined as the product amount produced per enzyme amount per unit reaction time. Unless otherwise stated, the reaction was conducted in 20 mM Kpi buffer (pH 7.0) containing 0.5 mg/mL enzyme at 40 °C for 10 min. To determine aldol condensation, the reaction was performed using 20 mM formaldehyde and 20 mM acetaldehyde. Then, 15 µL diluted samples and 50 µL 130 mM *O*-benzylhydroxylamine hydrochloride dissolved in 33:15:2 (*v/v/v*) pyridine: methanol: water were mixed and incubated at 25 °C for 2 h. The samples were diluted with 30 µL methanol, and 3-HPA was quantified using HPLC with 3-HPA as the standard. The kinetic parameters of DERA*_Tma_* were measured with different substrate concentrations. The *k_cat_* (min^−1^) and *K_M_* (mM) values were determined using non-linear regression with Michaelis–Menten equation using GraphPad Prism 8 software (GraphPad Software, San Diego, CA, USA).

### 3.7. Effects of pH and Temperature on DERA_Tma_ Activity

To evaluate the effect of pH on DERA*_Tma_* activity, pH values were varied from 5.5 to 9.0 using 20 mM MES buffer (pH 5.5–6.5), 20 mM Kpi buffer (pH 6.5–8.0), and 20 mM Tris buffer (pH 8.0–9.0) at 30 °C. The effect of temperature on DERA*_Tma_* activity was monitored as a function of time by incubating at 25, 30, 35, 40, 45, 50, 55, and 60 °C, and pH 7.0.

### 3.8. Effects of Enzyme and Substrate Concentrations on 3-HPA Production

To determine the optimal DERA*_Tma_* concentration for 3-HPA production from formaldehyde and acetaldehyde, 0.28–2.50 U/mL DERA*_Tma_* was incubated with 10 mM formaldehyde and 10 mM acetaldehyde. To determine the optimal substrate concentrations, the reactions were performed with 1.66 U/mL DERA*_Tma_* and 0–30 mM formaldehyde or acetaldehyde. Time-course reactions for 3-HPA production from formaldehyde and acetaldehyde were conducted with 1.66 U/mL DERA*_Tma_* for 180 min with 15 mM formaldehyde and 10 mM acetaldehyde.

### 3.9. Optimization of Reaction Conditions for One-Pot 3-HPA Production from C1 and C2 Compounds

To optimize the reaction conditions for 3-HPA production, the reactions were performed with 5 mM Mg^2+^, 3 mM NAD^+^, 1 M methanol, and 1 M ethanol in 20 mM Kpi buffer (pH 7.4). The effect of temperature on enzyme stability was monitored as a function of time by incubating at 25, 30, 35, 40, 45, 50, 55, 60, and 65 °C in 20 mM Kpi buffer (pH 7.4) for 120 min. To evaluate the effect of pH on aldol activity, pH values were varied from 6.0 to 9.0 using 20 mM Kpi buffer (pH 6.0–8.0) and 20 mM Tris–HCl buffer (pH 8.0–9.0) for 120 min at 45 °C. To determine the optimum DERA*_Tma_* and MDH*_Lx_* ratio, 0–2.22 U/mL DERA*_Tma_* was used, while 0.01 U/mL MDH*_Lx_* was used and 0–0.03 U/mL MDH*_Lx_* were used, while 1.39 U/mL DERA*_Tma_* was used at 45 °C for 120 min. To determine the optimal substrate concentrations, the reactions were performed in 20 mM Kpi buffer (pH 7.4) containing 1.39 U/mL DERA*_Tma_*, 0.026 U/mL MDH*_Lx_*, 5 mM Mg^2+^, and 3 mM NAD^+^ with 0–1 M methanol or ethanol for 120 min at 45 °C. Time-course reactions for one-pot 3-HPA production from methanol and ethanol were conducted in 20 mM Kpi buffer (pH 7.4) containing 1.39 U/mL DERA*_Tma_*, 0.026 U/mL MDH*_Lx_*, 5 mM Mg^2+^, 3 mM NAD^+^, 800 mM methanol, and 100 mM ethanol at 45 °C for 480 min.

### 3.10. Analytical Methods

Formaldehyde, acetaldehyde, and 3-HPA were quantified using HPLC equipped with a Gemini^®^ 5 µm C18 110 Å, LC column of 250 × 4.6 mm dimensions (Phenomenex, CA, USA) after derivatization as follows: 15 µL supernatant or reaction solution was mixed with *O*-benzylhydroxylaminehydrochloride solution (50 µL of 130 mM stock solution in pyridine:methanol:water 33:15:2). After being incubated at 25 °C for 2 h, the samples were diluted with 30 µL methanol and centrifuged at 8660× *g* for 3 min. The column was monitored by increasing the absorbance at 215 nm at 30 °C with a gradient of mobile phase A (water/trifluoroacetic acid, 100/0.1, *v/v*) and mobile phase B (acetonitrile/water/trifluoroacetic acid, 75/25/0.095, *v/v/v*) as follows: mobile phase B was increased from 10% to 100% for 30 min, decreased to 10% for 30–32 min, and stabilized for 32–50 min at 1 mL/min flow rate.

## 4. Conclusions

In this work, we identified the biosynthetic pathway of 3-HPA from formaldehyde and acetaldehyde using DERA*_Tma_*, which also catalyzes bioconversion of the cheap C1 and C2 chemicals, such as methanol and ethanol, to 3-HPA via the MDH*_Lx_* A164F cascade reaction. We optimized the reaction conditions, including pH, temperature, and enzyme and substrate concentrations, for DERA*_Tma_* activity and cascade reaction with MDH*_Lx_* A164F to improve enzymatic 3-HPA bioconversion. This is the first study on one-pot 3-HPA production from methanol and ethanol using two biocatalysts. In the future, the results of this study could be useful for industrial value-added product production from bio-renewable platform feedstocks via eco-friendly processes using biocatalysts.

## Figures and Tables

**Figure 1 ijms-23-03990-f001:**
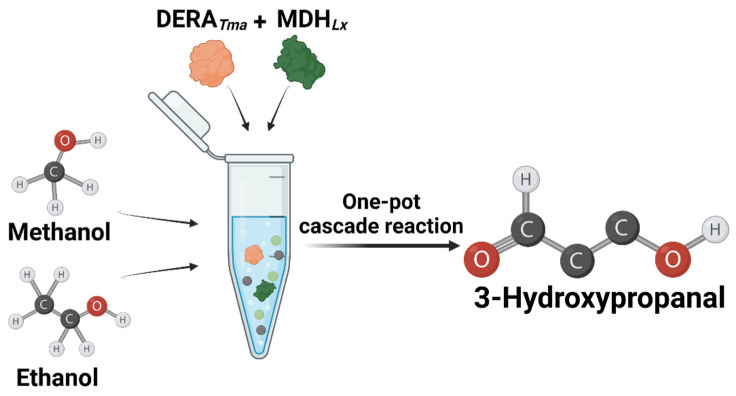
Scheme of the synthetic pathway of 3-hydroxypropanal from methanol and ethanol using DERA*_Tma_* and MDH*_Lx_*.

**Figure 2 ijms-23-03990-f002:**
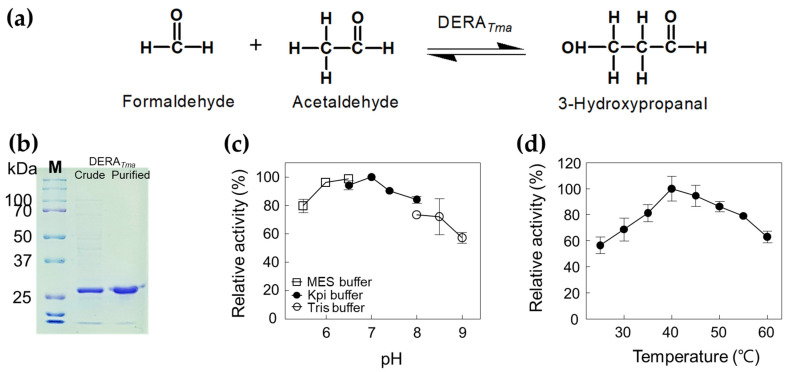
Characterization of DERA*_Tma_* for 3-HPA production from formaldehyde and acetaldehyde. (**a**) Scheme of 3-HPA production from formaldehyde and acetaldehyde by DERA*_Tma_*. (**b**) SDS-PAGE analysis of DERA*_Tma_*. All protein bands were stained using Coomassie blue. Molecular mass markers (M) indicate 250, 150, 100, 70, 50, 37, and 25 kDa. (**c**) Effect of pH on DERA*_Tma_* activity. The reactions were performed in 20 mM MES buffer (pH 5.5–6.5), 20 mM Kpi buffer (pH 6.5–8.0), and Tris buffer (pH 8.0–9.0) with 0.5 mg/mL enzyme, 20 mM formaldehyde, and 20 mM acetaldehyde at 30 °C for 10 min. (**d**) Effect of temperature on DERA*_Tma_* activity. The reactions were performed in 20 mM Kpi buffer (pH 7.0) with 0.5 mg/mL enzyme, 20 mM formaldehyde, and 20 mM acetaldehyde at different temperatures for 10 min. Data represent the means of two separate experiments, and error bars represent the standard deviation.

**Figure 3 ijms-23-03990-f003:**
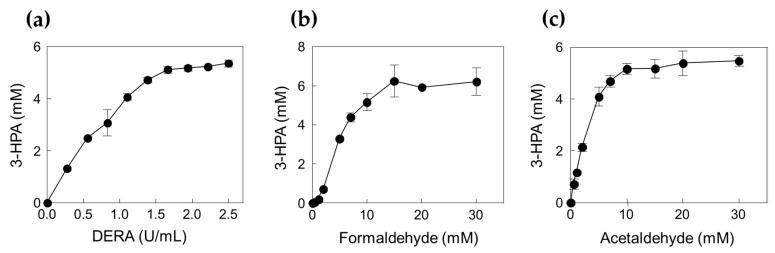
Effects of DERA*_Tma_* enzyme and substrate concentrations for the 3-HPA production. (**a**) Effect of DERA enzyme concentration. (**b**) Effect of formaldehyde concentration. (**c**) Effect of acetaldehyde concentration. The reactions were performed at 20 mM Kpi buffer (pH 7.0), DERA*_Tma_* enzyme, formaldehyde, and acetaldehyde at 40 °C for 120 min. Data represent the means of two separate experiments, and the error bars represent the standard deviation.

**Figure 4 ijms-23-03990-f004:**
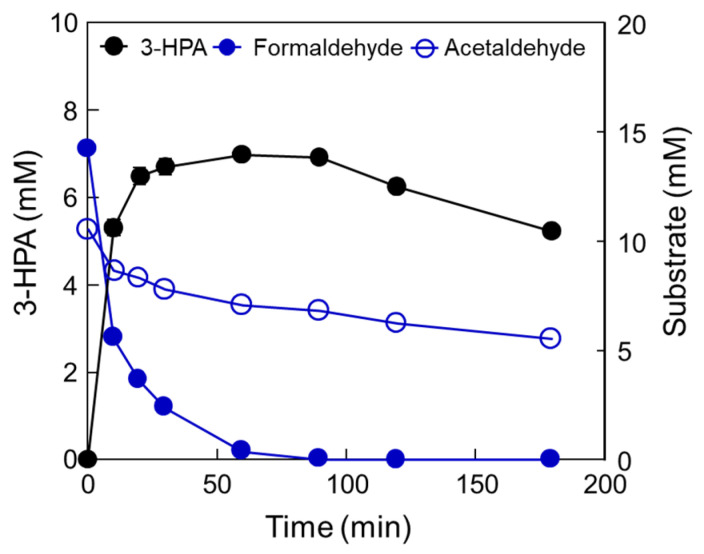
Production of 3-HPA from formaldehyde and acetaldehyde by DERA*_Tma_* under optimized conditions. Time-course reactions were performed in 20 mM Kpi buffer (pH 7.0) containing 1.66 U/mL enzyme with 15 mM formaldehyde and 10 mM acetaldehyde at 40 °C for 180 min. Data represent the means of two separate experiments, and error bars represent the standard deviation.

**Figure 5 ijms-23-03990-f005:**
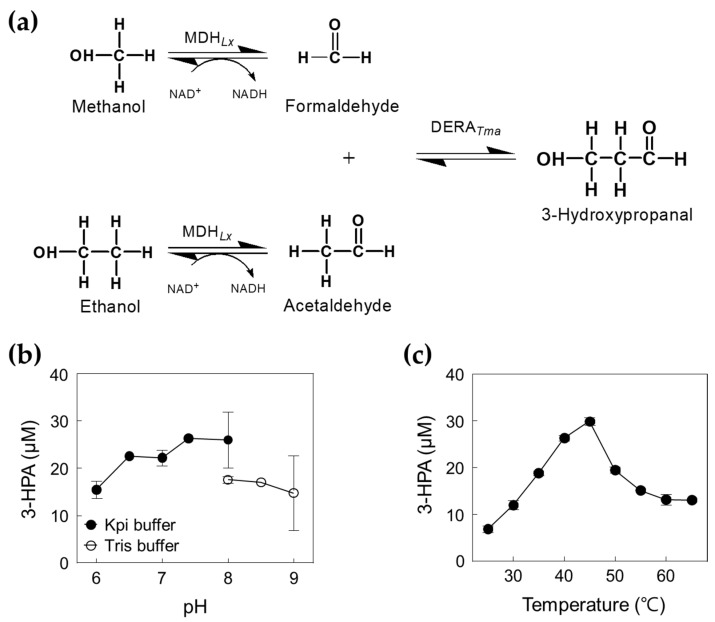
Production of 3-HPA from methanol and ethanol by DERA*_Tma_* and MDH*_Lx_*. (**a**) Scheme of 3-HPA production from methanol and ethanol by DERA*_Tma_* and MDH*_Lx_*. (**b**) Effect of pH for cascade reactions by DERA*_Tma_* and MDH*_Lx_*. The reactions were performed in 20 mM Kpi buffer (pH 6.0–8.0) and Tris buffer (pH 8.0–9.0) containing 2.22 U/mL DERA*_Tma_*, 0.01 U/mL MDH*_Lx_*, 1 M methanol, 1 M ethanol, 5 mM Mg^2+^, and 3 mM NAD^+^ at 45 °C for 120 min. (**c**) Effect of temperature for cascade reactions. The reactions were performed in 20 mM Kpi buffer (pH 7.4) containing 2.22 U/mL DERA*_Tma_*, 0.01 U/mL MDH*_Lx_*, 1 M methanol, 1 M ethanol, 5 mM Mg^2+^, and 3 mM NAD^+^ at different temperature ranges for 120 min. Data represent the means of two separate experiments, and the error bars represent the standard deviation.

**Figure 6 ijms-23-03990-f006:**
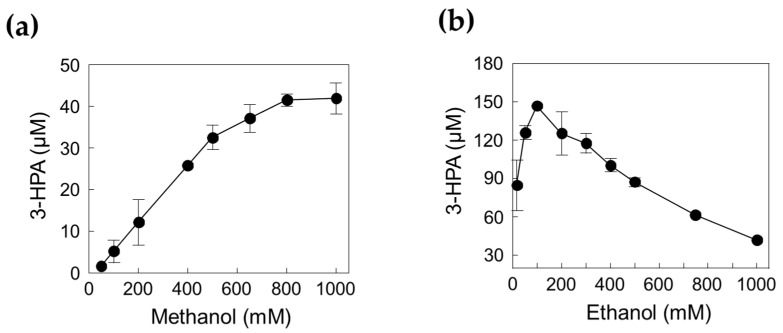
Effect of substrate concentration for 3-HPA production. (**a**) Effect of methanol. (**b**) Effect of ethanol. The reactions were performed in 20 mM Kpi buffer (pH 7.4) containing 1.39 U/mL DERA*_Tma_*, 0.026 U/mL MDH*_Lx_*, 5 mM Mg^2+^, and 3 mM NAD^+^ with methanol or ethanol at 45 °C for 120 min. Data represent the means of two separate experiments, and the error bars represent the standard deviation.

**Figure 7 ijms-23-03990-f007:**
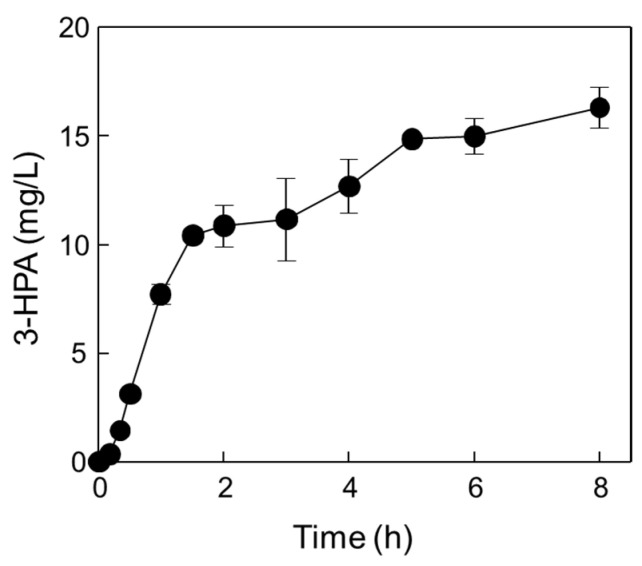
One-pot production of 3-HPA from methanol and ethanol by DERA*_Tma_* and MDH*_Lx_* under optimized reaction conditions. Time-course reactions were performed in 20 mM Kpi buffer (pH 7.4) containing 1.39 U/mL DERA*_Tma_*, 0.026 U/mL MDH*_Lx_*, 0.8 M methanol, 0.1 M ethanol, 5 mM Mg^2+^, and 3 mM NAD^+^ at 45 °C for 8 h. Data represent the means of two separate experiments, and error bars represent the standard deviation.

## Data Availability

The raw data supporting the conclusions of this article will be made available by the authors, without undue reservation, to any qualified researcher.

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
