# Peer review of "In Vitro One-Pot 3-Hydroxypropanal Production from Cheap C1 and C2 Compounds"

_ijms, 2022, doi:10.3390/ijms23073990_

Round 1

Reviewer 1 Report

 This is a good publication on enzymatic bio process of high value chemical compound.

Please could look at some small amelioration:

line 16 spelling propionic

figure 2 d: use white circle as in 2c for Kpi uffer could be helpfull for readers.

line 138 remove figure 3 and place Line 130 fig 3a, line 134 fig 3b and line135 fig 3c directly in the explanation. 

check if the standard deviation bars are in all the fif ie Fig 3a only one appearded!

figure 4: remove errors bars or add it on the figure.

figure 5: please put all the H on the scheme not only two for methanol, Ethanol and 3-hydroxypropanal

Reviewer 2 Report

It was a pleasure to review your article.

Please amend the following.

Line 100 Change to: 

The increasing order of pH values for the optimal activity...

Lines 105-106 Change to:

The increasing order of temperature values for the optimal activity... 

Line 342 change to:

...C2 chemicals, such as methanol and ethanol, to 3-HPA via the MDHLx A164F cascade reaction.

Reviewer 3 Report

Authors have presented the enzymatic method for 3-HPA production from inexpencive alcohol substrates (MeOH and EtOH). The method seems to be well optimized and documented and even the observed productivity is quite low, the results are pretty important. Manuscript is very well written and organized. I found only two small typos: -line 16 -> propioic -> correct: propionic and -line 385 -> Rsc -> capital: RSC
